# Phylogenomics investigation of sparids (Teleostei: Spariformes) using high-quality proteomes highlights the importance of taxon sampling

Paschalis Natsidis[1,2], Alexandros Tsakogiannis[1], Pavlos Pavlidis[3], Costas S. Tsigenopoulos [1]* & Tereza Manousaki[1]*

Sparidae (Teleostei: Spariformes) are a family of fish constituted by approximately 150 species with high popularity and commercial value, such as porgies and seabreams. Although the phylogeny of this family has been investigated multiple times, its position among other teleost groups remains ambiguous. Most studies have used a single or few genes to decipher the phylogenetic relationships of sparids. Here, we conducted a thorough phylogenomic analysis using five recently available Sparidae gene-sets and 26 high-quality, genome-predicted teleost proteomes. Our analysis suggested that Tetraodontiformes (puffer fish, sunfish) are the closest relatives to sparids than all other groups used. By analytically comparing this result to our own previous contradicting finding, we show that this discordance is not due to different orthology assignment algorithms; on the contrary, we prove that it is caused by the increased taxon sampling of the present study, outlining the great importance of this aspect in phylogenomic analyses in general.

[1] Institute of Marine Biology, Biotechnology and Aquaculture, Hellenic Centre for Marine Research, Heraklion, Greece. [2] School of Medicine, University of Crete, Heraklion, Greece. [3] Institute of Computer Science, Foundation for Research and Technology, Heraklion, Greece. *email: tsigeno@hcmr.gr; tereza@hcmr.gr

Teleostei represent the dominant group within ray-finned fish (Actinopterygii), with more than 26,000 extant species. Their evolution has been extensively studied through past decades, using a variety of data including fossil records, morphological characters and molecular data, leading to a gradual resolution of teleost phylogeny[1,2].

With the continuous emergence of new whole-genome sequences, phylogenomic techniques are applied to characterise the evolutionary relationships among species. Whole-genome information can help in resolving uncertain nodes, as well as provide stronger evidence on already established relationships. Regarding fish phylogeny, several genome-wide approaches have been implemented so far. One of the first efforts to study ray-finned fish phylogenomics was conducted by ref. [3]. Since then, several studies have been published using not only gene markers, but also non-coding elements such as the work of[4] who used UCE (ultra-conserved elements) to investigate the diversification of basal clades in ray-finned fish. Most genome papers include a phylogenomic analysis albeit with limited taxon sampling[5,6], while the use of whole-transcriptome data is being employed to uncover phylogenetic relationships of specific taxonomic groups as well[7,8]. With the emergence of new genomes and the possibilities of modern sequencing technologies, bigger datasets are becoming the norm. For example, a supermatrix of 1110 genes from 22 actinopterygians was assembled to resolve controversies regarding the evolution of the Otocephalan group[7]. Recently, the international project Transcriptomes of 1000 Fishes (Fish-T1K)[9] published a massive phylogenomic analysis including more than 300 fish species[10].

Sparidae (Teleostei: Spariformes), the focal family of this study, is a family of teleosts with increased commercial significance, constituted by ~150 species such as porgies and seabreams. The phylogenetic relationship of species within the family and among sparids and other teleost families has been tackled by multiple studies. However, most of them use satellite DNA or single gene markers with controversial findings (see ref. [11] for a review). Studies that focused on the relationships among sparids have reached various conclusions. Firstly, a close relation between the genera Pagrus and Pagellus has been proposed based on microsatellite DNA[12]. A few years later, de la Herrán et al.[13] presented a midpoint-rooted phylogeny of sparids using two microsatellite DNA families. In this tree, common pandora (Pagellus erythrinus) was placed closer to common dentex (Dentex dentex) rather than red porgy (Pagrus pagrus), a relationship also proposed by a recent tree including 1229 percomorphs from 23 concatenated genes[14]. In contrast to the aforementioned findings, two other studies placed common pandora together with the red porgy, leaving common dentex outside; the first study included 66 Sparidae species and 18 mitochondrial loci[15] and the second 91 Sparidae species and five loci[16]. This relationship is supported also by a recent single-gene approach using mitochondrial COI sequences from sparids inhabiting the Egyptian waters[17]. Thus, even though multiple studies have been conducted so far, the evolutionary relationship of Sparidae genera remains unclear.

The relationship of sparids to other fish families is another field of controversy. In the tree proposed by ref. [18], the sister clade of Sparidae contained four species from two different families (Lutjanidae & Haemulidae), however with the inclusion of only two loci from 48 species. More recent papers employed larger datasets such as a mitogenome data analysis from 75 teleosts[19] that placed Tetraodontiformes (puffer fish, sunfish) as the sister family of Sparidae, and another analysis using a six-loci supermatrix from 363 Mediterranean teleosts[20], which proposed Scarus ghobban (Family: Labridae) as the immediate sparid relative. Another tree of 44 actinopterygii mitogenome sequences placed two Lethrinidae (emperor fish) species, Lethrinus obsoletus and Monotaxis grandoculis, next to two sparids, Pagrus major and Spicara maena[21] Lethrinidae are also reported as the closest relatives of sparids in an investigation of Acanthomorpha (a subgroup of Teleostei) divergence times using a 10-gene dataset[22], and in the 1229-percomorph tree of Sanciancgo et al.[14]. A very recent and large-scale (303 fish species) phylogenomics study including four Sparidae transcriptomes (Evynnis cardinalis, Spondyliosoma cantharus, Acanthopagrus latus and Acanthopagrus schlegelii) presented a tree from 1105 loci that recovered the spinefoot Siganus guttatus (family: Siganidae) as the sister taxon to sparids, although with low support[10]. The testing of these last hypotheses using whole-genome information is not feasible yet due to lack of high-quality reference-based gene prediction of Lethrinidae or Siganidae genes.

Sparidae genetic data have been recently greatly enriched by transcriptomic studies[23,24] and two whole-genome sequencing datasets, those of gilthead seabream[25] and Chinese black porgy[26]. In ref. [25] using 2032 genes from 14 species, the large yellow croaker (Larimichthys crocea, family: Sciaenidae) and the European seabass (Dicentrarchus labrax, family: Moronidae) were placed as sister groups to gilthead seabream with high confidence. This scenario has been previously supported only by refs. [15,18].

Here, we revisited the phylogenetic relationship of sparids using newly obtained data from five genera with various reproductive modes, protogynous (Pagrus, Pagellus, Fig. 1a), protandrous (Sparus, Diplodus) and gonochoristic (Dentex). We built a phylogenomic dataset by including 26 non-sparid fish from a wide classification spectrum, focusing on high-quality, publicly available proteomes from species with sequenced genome, to ensure the highest data quality possible. The inferred phylogeny proposed Tetraodontiformes as closest relative to sparids than all teleost groups with a sequenced genome, a robust finding well supported by a variety of applied phylogenetic tests.

## Results

**Sparidae data preprocessing, taxon sampling and quality assessment.** The four Sparidae transcriptomes included 98,012 to 129,012 transcripts (Table 1). After keeping the longest ORF per gene, the largest set of sequences was that of common pandora, with 89,124 genes and the smallest that of red porgy with 62,116 genes. The gilthead seabream gene-set contained 61,850 genes.

Regarding the other teleost species, following a careful investigation of all the available sources for fish genomes, we formed a comprehensive dataset containing 31 species (Table 2). Apart from our five sparid gene-sets, we collected another 23 proteomes from NCBI, Ensembl and GigaDB databases, and three proteomes from other sources (species-specific databases, communication with paper authors). Almost half of all teleost fish with published whole-genome sequences[27] were included in the final dataset. Percomorphs (subdivision: Percomorphaceae) are well-represented in our dataset with 27 species, spanning seven out of the nine described series, as defined by ref. [2]. For the two unincluded series, Ophidiaria and Batrachoidiaria there is no available genome published up to now. Note that members of Salmonidae family with whole-genome sequence available (Salmo salar, Oncorhynchus mykiss) were not included in the final dataset because their extra whole-genome duplication[28] might hamper the orthology inference algorithms. Some species were excluded due to their low genome assembly statistics, such as the scaffold N50. The inclusion of multiple closely related species was avoided, for example we kept only one (Boleophthalmus pectinirostris, family: Gobiidae) out of the four available mudskipper genomes[29] using the assembly statistics as selection criterion. Apart from the 27 percomorphs, the remaining four species of the final dataset were the Paracanthopterygii member

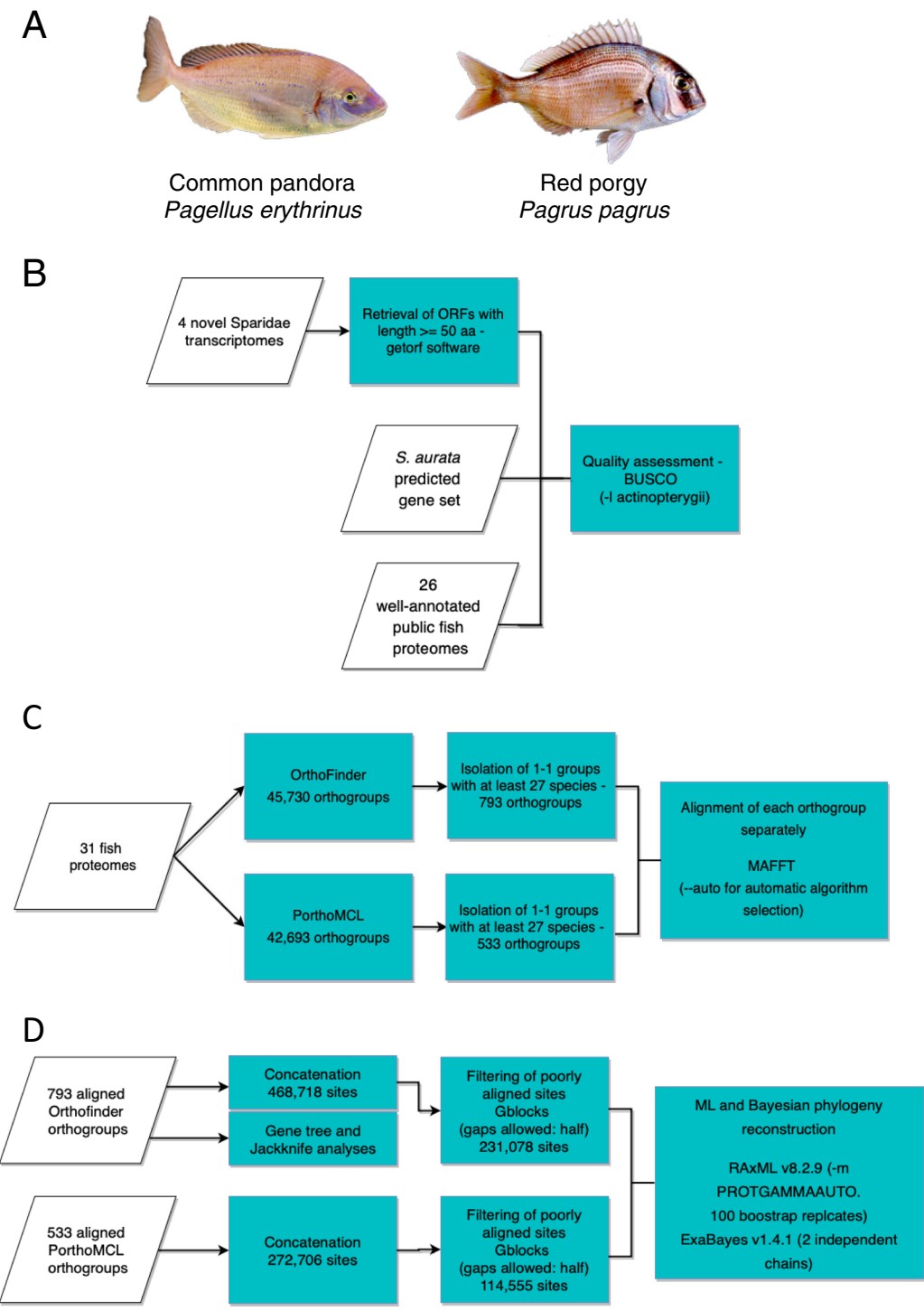

**Fig. 1 a** The two protogynous species used in this study, *Pagellus erythrinus* and *Pagrus pagrus*. The other three Sparidae used in this study are gonochoristic (*Dentex dentex*), obligatory protandrous (*Sparus aurata*) and rudimentary protandrous (*Diplodus puntazzo*). Image copyrights: Alexandros Tsakogiannis.
**b–d** The main workflow divided into three main components: Taxon sampling and quality assessment, Orthology assignment and MSA, Phylogenomics analysis respectively

Atlantic cod (*Gadus morhua*, order: Gadiformes), two members of the Ostariophysi superorder, zebrafish (*Danio rerio*, order: Cypriniformes) and the blind cavefish (*Astyanax mexicanus*, order: Characiformes) and the Holostei spotted gar (*Lepisosteus oculatus*) as an outgroup.

To assess the quality of each gene-set we ran BUSCO analysis. The results showed that the sharpsnout seabream has the lowest number of BUSCO sequences with 3347 (73%) out of the 4584 genes (Fig. 2, Supplementary Table 1). The other sparid gene-sets scored higher BUSCO statistics, outperforming even some of the 26 datasets from online sources. The common dentex dataset contained 3,876 (84.5%) BUSCO genes, common pandora had 3,954 (86.3%), while red porgy had 3,945 (86.1%) genes. The geneset of gilthead seabream contained 3,910 (85.3%) genes and

**Table 1 Preprocessing of the four Sparidae transcriptomes**

| Species | Transcripts in assemblies | Total number of ORFs found with length >50 a.a. | Transcripts with at least one ORF (% of transcripts with ORF) | Number of coding genes used in the final analysis |
|---|---|---|---|---|
| *D. puntazzo* | 129,012 | 1,272,493 | 113,208 (87.75%) | 83,527 |
| *D. dentex* | 118,258 | 1,285,298 | 113,684 (96.13%) | 78,451 |
| *P. erythrinus* | 141,309 | 1,416,980 | 129,523 (91.66%) | 89,124 |
| *P. pagrus* | 98,012 | 1,264,706 | 91,787 (93.64%) | 62,116 |

For each transcriptome we present the number of sequences contained, the number of open reading frames (ORF) found, the number of transcripts with at least one ORF and the final proteome included in the analysis after keeping the longest ORF per gene

**Table 2 List of species included in the phylogenomic analysis**

| Species | Series (for Percomorphaceae) | Source | Reference | #of proteins |
|---|---|---|---|---|
| *A. mexicanus* | (Ostariophysi) | Ensembl database | [61] | 22,998 |
| *B. pectinirostris* | Gobiaria | NCBI ftp server | [29] | 21,541 |
| *C. argus* | Anabantaria | GigaDB | [6] | 20,541 |
| *C. semilaevis* | Carangaria | NCBI ftp server | [62] | 24,489 |
| *D. rerio* | (Ostariophysi) | Ensembl database | [63] | 25,644 |
| *D. dentex* | Eupercaria | in-house sequenced | PRJNA481721 | 83,527 |
| *D. labrax* | Eupercaria | species database | [64] | 26,719 |
| *D. puntazzo* | Eupercaria | in-house sequenced | [24] | 78,451 |
| *G. morhua* | (Paracanthopterygii) | Ensembl database | [65] | 19,978 |
| *G. aculeatus* | Eupercaria | Ensembl database | [66] | 20,625 |
| *H. erectus* | Syngnatharia | GigaDB | [67] | 20,788 |
| *K. marmoratus* | Ovalentaria | NCBI ftp server | [68] | 25,257 |
| *L. crocea* | Eupercaria | NCBI ftp server | [69] | 28.009 |
| *L. calcarifer* | Carangaria | NCBI ftp server | [5] | 22,221 |
| *L. oculatus* | (Holostei) | Ensembl database | [70] | 18,304 |
| *M. peelii* | Eupercaria | GigaDB | [71] | 26,539 |
| *M. mola* | Eupercaria | GigaDB | [72] | 19,605 |
| *M. albus* | Anabantaria | NCBI ftp server | [73] | 24,943 |
| *N. coriiceps* | Eupercaria | NCBI ftp server | [74] | 25,937 |
| *O. niloticus* | Ovalentaria | Ensembl database | [75] | 21,383 |
| *O. latipes* | Ovalentaria | Ensembl database | [76] | 19,603 |
| *P. erythrinus* | Eupercaria | in-house sequenced | [23] | 89,124 |
| *P. pagrus* | Eupercaria | in-house sequenced | [23] | 62,116 |
| *P. charcoti* | Eupercaria | provided by authors | [34] | 32,713 |
| *P. formosa* | Ovalentaria | Ensembl database | [77] | 23,315 |
| *S. dumerili* | Carangaria | NCBI ftp server | Araki et al., unpublished | 24,000 |
| *S. aurata* | Eupercaria | in-house sequenced | [25] | 61,850 |
| *T. rubripes* | Eupercaria | Ensembl database | [78] | 18,433 |
| *T. nigrovirdis* | Eupercaria | Ensembl database | [79] | 19,511 |
| *T. orientalis* | Pelagiaria | species database | [80] | 26,433 |
| *X. maculatus* | Ovalentaria | Ensembl database | [81] | 20,343 |

For each species we indicate the series (or another distinct taxonomic group for the non-Percomorphaceae) they belong to, the sources of the proteomes used, the reference paper and the number of the protein sequences contained in each proteome

had the fewest missing genes among the five sparids. As for the publicly available proteomes, the ones downloaded from the Ensembl database presented the smallest number of missing genes (from <10–100), while datasets obtained from NCBI contained more duplicated genes.

**Orthology assignment and superalignments construction.** The total number of genes from all 31 proteomes included in the orthology assignment analysis was 974,940. OrthoFinder and PorthoMCL identified 45,730 and 42,693 groups of orthologous genes, respectively (Table 3). Following filtering, we kept 793 and 533 groups from each dataset to construct the two super-alignments. The superalignment of OrthoFinder groups consisted of 468,718 amino acids and the one of PorthoMCL groups of 321,695. Gblocks filtering retained 231,078 (49%) and 141,608 (44%) sites, respectively.

**Phylogenomic analysis.** All RAxML runs selected JTT[30] as the model of evolution that best explains our dataset, with gamma distribution on rates and empirical base frequencies (noted as PROTGAMMAJTTF). Maximum likelihood trees for both OrthoFinder (Fig. 3) and PorthoMCL (Supplementary Fig. 1) superalignments resulted in similar topologies for most species. Firstly, they agreed on the monophyly of the five sparid species. The common pandora and the red porgy were grouped together, with common dentex as their closest relative. The gilthead seabream and the sharpsnout seabream were placed together in the clade that diverged first within the sparid lineage. All intra-familial relationships of Sparidae were supported by a 100 bootstrap value in both OrthoFinder and PorthoMCL maximum likelihood trees.

The closest group to sparids recovered was Tetraodontiformes. *Takifugu rubripes* (green spotted puffer, family: Tetraodontidae)

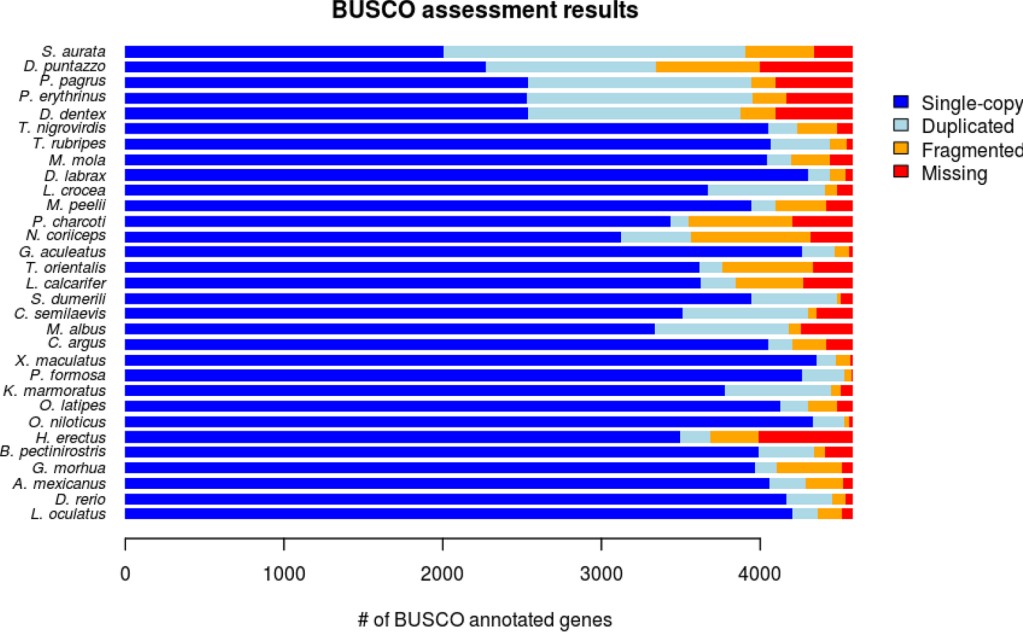

**Fig. 2** Quality assessment using BUSCO. The five Sparidae proteomes are shown in the top five bars

| Table 3 Comparison of the two orthology inference tools and the respective superalignments | | | | | |
|---|---|---|---|---|---|
| **Software** | **Groups of orthologs returned** | **Single-copy groups with at least 27 taxa** | **Average aligned group length (a.a.)** | **Concatenated alignment length (a. a.)** | **Filtered alignment length (a.a.)** |
| OrthoFinder | 45,730 | 793 | 591.06 | 468,718 | 231,078 |
| PorthoMCL | 42,693 | 533 | 603.56 | 321,695 | 141,608 |
| OrthoFinder provided greater number of orthogroups than PorthoMCL both initially and after filtering for 1-1 groups with representation from at least 27 species | | | | | |

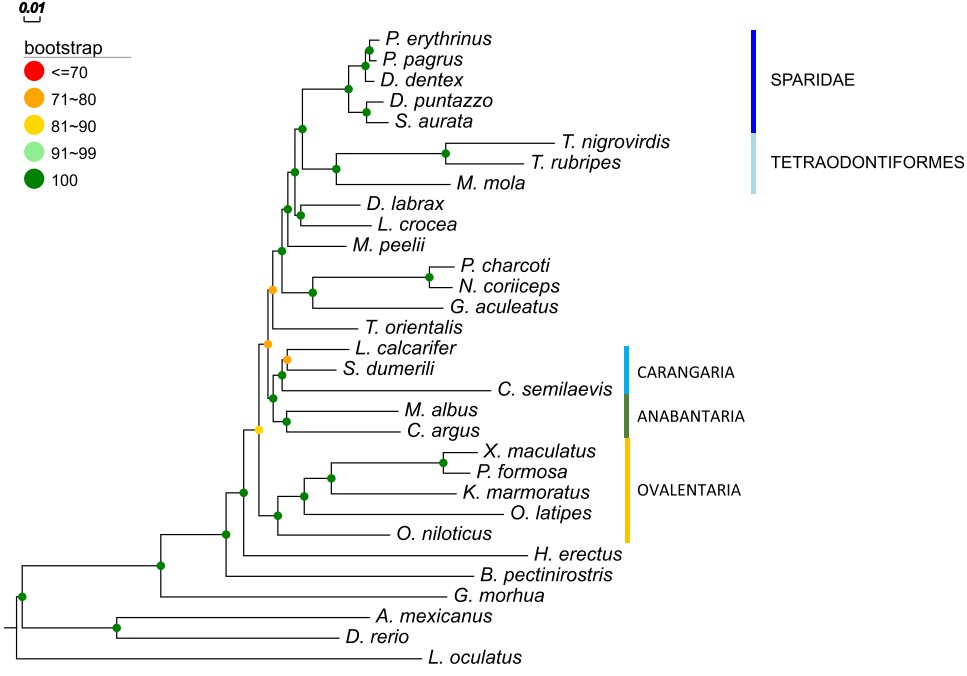

**Fig. 3** Maximum likelihood (RAxML) tree of 793 concatenated OrthoFinder groups using JTT + F + Γ model and 100 bootstrap replicates. The spotted gar (*L. oculatus*) was used as an outgroup

and *Tetraodon nigroviridis* (Japanese puffer, family: Tetraodontidae) exhibited longer branch lengths than the third Tetraodontiformes member, the *Mola mola* (ocean sunfish, family: Molidae). RAxML tree searches using different subsets of Tetraodontiformes and Sparidae taxa (Supplementary Fig. 2) agreed on their proposed relationship, with maximum bootstrap support at all times. *Larimichthys crocea* (large yellow croaker, family: Sciaenidae) and the *Dicentrarchus labrax* (European sea bass, family: Sciaenidae) were grouped together, as the immediately closest group to the Sparidae/Tetraodontiformes clade.

The two species that reside in the Antarctic waters, *Parachaenichthys charcoti* (Antarctic dragonfish, Family: Bathydraconidae) and *Notothenia coriiceps* (Antarctic bullhead, Family: Nototheniidae) were placed in the same clade, with *Gasterosteus aculeatus* (stickleback, Family: Gasterosteidae) as their closest relative. These three fish are all members of the order Perciformes[2].

The OrthoFinder tree had maximum bootstrap support values (100) assigned in all nodes of the above findings, that describe the phylogenetic relationships of the 14 Eupercaria (Eu) fish of our dataset. PorthoMCL tree recovered identical topology for the Eu, with all nodes presenting maximum bootstrap support values, except from the croaker/seabass ancestor (93).

The monophyly of each of the Carangaria (C), Anabantaria (A) and Ovalentaria (O) series was supported by both OrthoFinder and PorthoMCL maximum likelihood trees with high intra-series support values. However, the inter-series relationships of these three groups recovered by the two trees do not agree. OrthoFinder tree suggested the grouping of C/A cluster together with the Eu, while PorthoMCL tree placed C/A and O in the same clade, although with low support (bootstrap value 49).

Another point of discordance between the two maximum likelihood trees was the position of the *Thunnus orientalis* (pacific bluefin tuna, family: Scombridae), which is a member of the Pelagiaria series[2]. In the OrthoFinder tree, tuna was placed next to the Eu clade, while in PorthoMCL tree it is placed outside the Eu/C/A/O cluster. Both of these placements were supported by a not so high bootstrap proportion (73 and 71 respectively).

For the non-percomorph fish, the two maximum likelihood trees converged on grouping the two Ostariophysi members, *Danio rerio* and *Astyanax mexicanus*, together. These two fishes diverged first from the rest of teleosts, with the next divergence giving the Atlantic cod clade, followed by the mudskipper. All nodes described were assigned maximum bootstrap value in both OrthoFinder and PorthoMCL trees.

To summarize the information of the 100 OrthoFinder and the 100 PorthoMCL jackknifed tree datasets, we built two consensus trees. The consensus tree on the OrthoFinder jackknifes (Supplementary Fig. 3a) presented identical topology with the main OrthoFinder RAxML tree. Support for the controversial nodes (BS < 85) of the main tree was increased to 93, except for the tongue sole split that was present in 83 out of the 100 jackknifed trees. The consensus tree of the 100 PorthoMCL jackknifed trees (Supplementary Fig. 3b) presented identical topology with the main PorthoMCL RAxML tree. The controversial nodes (BS < 85) of the main tree maintained their low support in the jackknife consensus tree as well. However, the split of common pandora and red porgy received a support value of 79 in the jackknife consensus tree, while it was recovered with 100 bootstrap support on the main PorthoMCL tree.

The two consensus trees from OrthoFinder and PorthoMCL Bayesian analyses recovered identical topologies, except for the relationships among the three Carangaria species. The OrthoFinder tree (Supplementary Fig. 4a) proposed the grouping of *Lates calcarifer* (Asian seabass, family: Latidae) and *Seriola dumerili* (greater amberjack, family: Carangidae) group leaving the *Cynoglossus semilaevis* (tongue sole, family: Cynoglossidae) outside, while the PorthoMCL tree (Supplementary Figs. 4b) grouped the tongue sole together with the greater amberjack. The tongue sole was assigned a longer branch than its two relatives. Both Bayesian trees presented posterior probabilities equal to 1.0 in all of their nodes.

To identify any possibly rogue taxa, we ran RogueNaRok on the bootstrap replicates of each maximum likelihood tree search. The results did not drop any taxa as rogue, with RBIC scores calculated at 0.966 and 0.939 for OrthoFinder and PorthoMCL maximum likelihood trees, respectively. Nevertheless, we tested how the removal of some possibly ambiguous taxa affected the topology and the support values. For this analysis, we used the 793 orthogroups of OrthoFinder.

To check how the long branch of the tongue sole affected the proposed phylogeny, we discarded its sequences from all OrthoFinder groups and built maximum likelihood tree anew. This tree suggested identical topology to the one with all 31 species, but with a slight increase of the bootstrap support values (Supplementary Fig. 5a).

We also examined whether the pacific bluefin tuna dataset is related to the low bootstrap support values of the tree. To that end, we furtherly reduced the OrthoFinder groups to 29 species by removing tuna sequences as well. The resulting trees proposed the same topology as the initial trees for the remaining species, but this time with all nodes at maximum support value (Supplementary Fig. 5b).

To check how our result compares to our previous work, the tree suggested in the gilthead seabream genome paper, we used CONSEL. The results strongly supported the topology with Tetraodontiformes as most closely related group to sparids, as opposed to the topology of ref. [25], that suggested the croaker/seabass clade as closest one to sparids. The p-values of all tests were equal to 1 (Table 4) for both OrthoFinder and PorthoMCL datasets. Specifically, for the approximate unbiased (au) test, which is the main result of a CONSEL run, we may reject the possibility that a tree is the most likely tree among all candidates when AU < 0.05 at the significance level 0.05. Thus, AU = 1.0 provides very strong evidence for Tetraodontiformes against croaker/seabass as the closest group to sparids, based on this dataset.

To test whether the selection of the orthology assignment algorithm is responsible for the discordance between the present study and the gilthead seabream genome paper, we ran OrthoFinder and PorthoMCL using the same 14 species as we have done in ref. [25]. After filtering for single-copy groups with maximum one species missing, we were left with 2192 and 1366 genes for the two tools respectively. After concatenating them into two separate superalignments and filtering the ambiguous sites, we conducted a maximum likelihood tree estimation for each dataset. The two resulting trees were identical both between each other, and with the tree presented by ref. [25]. The European seabass and the large yellow croaker were placed as sister taxa to the gilthead seabream, while the two puffer fish were recovered as immediate relatives to these three fish.

**Gene tree incongruence**. To assess the (in)congruence between the gene trees and the estimated species trees, we constructed individual trees for the groups of orthologs that contained sequence information from all 31 species. Hundred and thirty five OrthoFinder and 78 PorthoMCL groups satisfied the above criterion, and their trees were used to build a consensus tree, and to calculate internode certainty (IC and ICA) and tree certainty (TC and TCA) values, related to the corresponding species tree. The two consensus trees contained multiple multifurcating nodes

**Table 4 Comparison of the topology presented here, with Tetraodontiformes as closest group to Sparidae, and the topology suggested in[25], with croaker and seabass as closest group to Sparidae, using CONSEL**

| Tree | obs | au | np | bp | kh | sh | wkh | wsh |
|---|---|---|---|---|---|---|---|---|
| OrthoFinder | | | | | | | | |
| Nats | −558.7 | 1.000 | 1.000 | 1.000 | 1.000 | 1.000 | 1.000 | 1.000 |
| Paul | 558.7 | 4e-07 | 2e-06 | 0 | 0 | 0 | 0 | 0 |
| PorthoMCL | | | | | | | | |
| Nats | −345.8 | 1.000 | 1.000 | 1.000 | 1.000 | 1.000 | 1.000 | 1.000 |
| Paul | 345.8 | 1e-50 | 2e-17 | 0 | 0 | 0 | 0 | 0 |

The table shows the p-values of various statistical tests. We may reject the possibility that a topology is the most likely to be the true when au < 0.05 at the significance level 0.05[55]. Nats: present study; Paul[25], in press
*obs* observed log-likelihood difference, *au* approximately unbiased test, *np* multiscale bootstrap probability, *bp* usual bootstrap probability, *kh* Kishino-Hasegawa test, *sh* Shimodaira-Hasegawa test, *wkh* weighted Kishino-Hasegawa test, *wsh* weighted Shimodaira-Hasegawa test

(Supplementary Fig. 6a, b). The results of IC/TC analysis suggested low conflict in shallow nodes of the trees, i.e., at family level (Supplementary Fig. 6c, d) compared to deeper nodes. For example, the Sparidae monophyly was highly supported (IC = 0.865 in the OrthoFinder dataset). However, high conflict was observed in deeper divergences, with even negative IC values at some ambiguous nodes such as the tuna placement. Negative IC values show that the most represented topology within the gene trees is not the one recovered in the reference tree. Relative TC values were reported to be 0.295 and 0.212 for OrthoFinder and PorthoMCL datasets respectively.

## Discussion

Here we analysed a teleost phylogenomic dataset and questioned the position of Sparids within the tree of teleosts using high-quality gene prediction datasets. Our results suggested Tetraodontiformes as the closest group to Sparidae and grouped the analysed sparids in accordance to their known relationships.

Regarding within-sparids relationships, all trees that were built in the present analysis recovered a single topology (Fig. 3) for the five species used. The resulted topology agrees with previous studies[15–17]. Interestingly, the species with the same reproductive mode were grouped together, i.e. red porgy and common pandora are protogynous, the protandrous gilthead seabream is grouped with the rudimentary protandrous sharpsnout seabream and the gonochoristic common dentex falls in between the two groups. In general, the members of Sparidae family exhibit a variety of reproduction modes[31] and further investigation is needed about the evolutionary origins of these different patterns.

As for the relationships of sparids with other teleost groups, our results showed that, from the species included in the analysis, Tetraodontiformes is the closest group to sparids. This has been frequently reported in the literature as well[19,20]. However, our earlier phylogenomics study presented in the gilthead seabream genome paper[25], the first thorough analysis including a Sparidae species and 14 other taxa, proposed with high confidence the yellow croaker and the European seabass as more closely related to sparids than the Tetraodontiformes. To understand why the two phylogenomic analyses find such controversial results, we tracked down the main differences of the present work to ref. [25]. The main differences are: the algorithm used for identifying the orthology groups, and the denser taxon sampling of the present study. Regarding the first, the groups of ortholog genes in ref. [25] were recovered using the OMA standalone[32], a software considered to have high specificity, but low sensitivity in finding the true orthologous clusters[33]. To see whether the selection of orthology inference algorithm affected the resulted phylogeny, we repeated our orthology assignment employing OrthoFinder and PorthoMCL using only the 14 taxa used in ref. [25], and conducted

the phylogenomic analysis. Interestingly, the analysis of this reduced dataset (Supplementary Fig. 7) was in total agreement with the one we reported in ref. [25]. This suggests that the discordance with the present paper results is not due to the selection of different orthology inference algorithm, and might be explained by the more ample taxon sampling of the present study, both within sparids (5 vs 1 by ref. [25]) but also in the rest of teleost taxa (26 vs 14 by ref. [25]). Another hypothesis potentially explaining the discordance of the two analyses could be that in this study we included a third species of Tetraodontiformes, the ocean sunfish, that might have overcome a potential long branch attraction present in ref. [25] tree. To test this hypothesis, we removed the ocean sunfish from our 31-species dataset and rebuilt tree and Tetraodontiformes remained as the closest group (Supplementary Fig. 2b). This result remained the same even when we reran the analysis using only gilthead seabream from the five sparids (Supplementary Fig. 2a). Finally, CONSEL analysis, given our superalignment, strongly supported the present analysis topology against the one of ref. [25]. All these pieces of evidence corroborate the robustness of the results presented here and at the same time underline how critical dense taxon sampling is.

The positioning of the non-sparid teleosts in our trees arose a noteworthy issue as well. Our OrthoFinder tree placed tuna as sister taxon to the Eupercaria clade, while the PorthoMCL tree proposed that tuna diverged right after the seahorse divergence. Both trees assigned relatively low support on the tuna split and some other nodes close to it. When we removed tuna sequences from the dataset, all support values of the trees increased to 100. Resolving the position of tuna within the fish phylogeny has been an object of contradiction in the existing literature. In the tree of ref. [20], the Scombriformes order was placed very close to the Gobiiformes, the order that mudskippers belong into. However, the 1410-species review of[1] grouped together the orders of Scombriformes and Syngnathiformes, suggesting a closer relationship of tunas to seahorses, rather than mudskippers. This relationship was confirmed by refs. [2,14], that proposed Syngnatharia, the series of seahorses, as closest relatives of Pelagiaria, the series of tunas. In both studies though, the Syngnatharia/Pelagiaria branch was assigned a moderate support value (<89). Only very recently, the relationship between seahorses and tunas was recovered with high confidence[10]. This relationship remains to be confirmed by future studies.

Apart from the tuna positioning, most of our other findings on phylogenetic placement of the non-sparid fish are in agreement with the existing literature. Indicatively, the two Antarctic fishes (dragonfish and bullhead) and the stickleback were placed in the same clade in our study. This is in agreement with the results presented in the dragonfish genome paper[34] and the study of ref. [10] as well.

Technically speaking, taxon sampling is a crucial part of a phylogenetic analysis. This has been shown by multiple studies (e.g. refs. [35–37]) and by a review tackling the impact of taxon sampling on phylogenetic inference[38]. Incongruence in molecular phylogenies can also be resolved by increasing the number of genes included in the analysis[39]. Therefore, it is necessary for any phylogenomic analysis to include as many taxa as possible, without reducing the amount and the quality of the loci used to build the tree. In our case, we have a much denser taxon sampling compared to[25] but reduced number of genes, which is normal when taxa inclusion increases. However, we recovered phylogeny of ref. [25] even with our dataset when keeping only the species used in that study. Thus, in this case it seems that although in ref. [25] we used many more genes the taxon sampling was the crucial factor.

Another important factor in phylogenomic studies is the selection of the orthology inference algorithm. Various issues for most tools, regarding computational time and accuracy have been described and reviewed by ref. [40]; however, recently developed promising tools improve greatly the orthology inference in both of the above aspects. Here, we chose to employ two recently developed graph-based orthology inference software tools, OrthoFinder and PorthoMCL. OrthoFinder has improved accuracy compared to other algorithms, while PorthoMCL is a faster implementation of the OrthoMCL algorithm. They both use BLAST results to infer orthology, but OrthoFinder steps include a normalization of the BLAST bit scores according to the length of the genes[41]. This normalization solves a previously unaddressed bias that favoured longer genes, as they were assigned greater bit scores. In our case, this led to an increased number of orthogroups returned by OrthoFinder both initially (45,730 vs 42,693 in PorthoMCL) and after filtering for 1–1 groups with at least 27 taxa (793 vs 533 in PorthoMCL). The average length of these groups, however, was slightly smaller in the OrthoFinder groups (591.06 sites/group vs 603.56 in PorthoMCL). Moreover, the results of the jackknifed trees analysis suggested that OrthoFinder groups were more robust, and a tree with 70% of them at random will most likely recover the topology of the whole dataset. On the other hand, a random 70% of the PorthoMCL groups was not always enough to fully recover the relationship between common pandora and red porgy, as well as some of the deeper splits.

Gene tree analysis was unable to recover the topologies that resulted from the supermatrix approach, suggesting that phylogenetic signal in gene trees is inadequate. The consensus trees for both OrthoFinder and PorthoMCL 31-species gene trees presented mostly polytomies, while IC/ICA values were low, even negative, in some deeper nodes. The amount of discordance among the gene trees, as well as the conflict between the gene trees and the species tree recovered via supermatrix approach indicates that this type of analysis is not suitable for our dataset. This is an innate property in cases where gene trees are used to infer species phylogeny[42].

## Methods

**Sparidae data preprocessing, taxon sampling and quality assessment**. The transcriptomes from brains and gonads of common dentex (*Dentex dentex*), sharpsnout seabream (*Diplodus puntazzo*), common pandora (*Pagellus erythrinus*) and red porgy (*Pagrus pagrus*) were obtained from our previous studies[23,24]. We processed the four sparid transcriptomes using the EMBOSS v6.6.0.0 software getorf[43] with the option '-minsize 150', to recover all open reading frames (ORFs) of length ≥ 50 amino acids. The longest ORF was kept for each gene using a Python script. For gilthead seabream (*Sparus aurata*), we obtained the full gene-set from its genome sequence publication[25].

The selection of non-sparid taxa included in the analysis was based on: the availability of a well-annotated predicted gene-set, the availability of a genome paper that describes an elaborated gene prediction pipeline, and representation of a wide range of the different teleost groups. Our selection process resulted to the inclusion of 26 fish proteomes, recovered mainly from NCBI[44], Ensembl[45] and GigaDB[46] databases (Table 2), in addition to the five sparids, forming a 31 taxa dataset (Fig. 1b). We selected the spotted gar (*Lepisosteus oculatus*), a member of Holostei, as an outgroup for our analysis. Most of the fishes dismissed from the analysis but with a genome sequence had either low assembly statistics, or their inclusion was redundant, since other closely related taxa were selected. This was done in order to reduce computational costs. Proteomes retrieved from NCBI and Ensembl databases had multiple isoforms for some genes. We processed those proteomes to keep the longest isoform per gene using in-house scripting.

To assess the quality of the retrieved gene-sets, we employed the BUSCO v3[47] software, using the '-l actinopterygii' option to enable the proper lineage library for our data (Fig. 1b). This library consisted of 4584 genes that were expected to be present in at least 90% of the species in the actinopterygii lineage. So, a high representation of the BUSCO genes in each of our datasets is an indicator of quality and completeness of the gene-sets. BUSCO provided statistics for genes found in complete form, fragmented or duplicated in the tested datasets.

**Orthology assignment and superalignments construction**. To investigate orthology relationships among the sparid gene-sets and the downloaded proteomes, we employed two different tools (Fig. 1c), OrthoFinder v2.1.2[41] and PorthoMCL[48]. The OrthoMCL algorithm[49] uses Markov clustering to group (putative) orthologs and paralogs. PorthoMCL is a parallel implementation of the OrthoMCL algorithm, making genome-scale orthology assignment computationally feasible. The OrthoFinder algorithm solves a previously untackled gene length bias in orthogroup inference, by normalising the BLAST bit scores.

We discarded all ortholog groups with more than one gene per species to avoid potential paralogies. From the resulted single-copy groups, we collected those with representation of at least 27 of the 31 taxa, so that every group contained at least 1 of the 5 sparids. We used Python scripting to retrieve the amino acid sequences of each orthogroup and used them for downstream analyses.

To align the sequences of each orthogroup separately we employed MAFFT v7[50], allowing the–auto parameter to determine the most suitable alignment method. The alignments of OrthoFinder and PorthoMCL groups were concatenated into two distinct superalignments using a Python script (Fig. 1d). The superalignments were then filtered with Gblocks v0.91b[51] to remove poorly aligned sites, changing the parameter Allowed Gap Positions to half and leaving all other parameters at default values.

**Phylogenomic analysis**. Each of the two obtained filtered superalignments, one from OrthoFinder groups and one from PorthoMCL groups, was provided as input to RAxML v8.2.9[52] to search for the maximum likelihood tree. The parameter -m PROTGAMMAAUTO was selected to automatically select the model that best fits our dataset. One hundred rapid bootstrap replicates were drawn from the input alignment during each RAxML run. Apart from bootstrap resampling, we ran maximum likelihood on 100 jackknifed datasets. For that, a random 30% of the orthogroups was excluded each time from the supermatrix, keeping the rest 70% of the orthogroups. The random split was achieved using bash and Python scripts. A majority rule consensus tree was built to summarize the bipartition information of the 100 jackknifed trees using the RAxML -J MRE option.

Bayesian Inference was performed using ExaBayes v1.4.1[53]. Two independent chains were initiated in parallel using the -R 2 option. The Markov chain Monte Carlo (MCMC) sampling of trees was automatically stopped after 1,000,000 generations due to convergence of the two chains, after discarding the default 25% burn-in. We inspected the sampled distributions of the parameters and confirmed the sufficiency of the effective sample sizes (ESS > 200) of all sampled parameters with the postProcParam utility. Finally, we built a consensus tree from the two sets of trees using the consense utility of ExaBayes.

To identify potential rogue taxa in the 100 bootstrap replicates of the two maximum likelihood trees we employed RogueNaRok v1.0[54]. The RogueNaRok algorithm optimizes the relative bipartition information criterion (RBIC), which is defined as the sum of all support values divided by the maximum possible support in a fully bifurcating tree with the initial (i.e., before pruning any rogues) set of taxa. The algorithm prunes taxa until RBIC cannot be further improved.

To compare the placement of Sparidae in present trees with the one we conducted in the gilthead seabream genome paper we employed CONSEL v0.20[55]. CONSEL calculates *p*-values for various statistical tests based on the per-site log likelihoods for the candidate trees given a sequence alignment. These tests include the approximate unbiased (AU) test[56], the K-H test[57] and the S-H[58] test among others. The output of CONSEL allows to determine which of the candidate topologies is most likely to be the true one given the data. We obtained the per-site log likelihoods using the RAxML option '-f G'. We applied CONSEL analysis on both OrthoFinder and PorthoMCL datasets.

We also tested if the aforementioned discordance is due to the selection of the orthology assignment algorithm. To that end, we kept only the 14 species that we used in ref. [25] and ran OrthoFinder and PorthoMCL anew. We then selected single-copy groups with at least 13 of the 14 species and aligned them using MAFFT. The alignments of each tool were separately concatenated into two superalignments using custom Python scripts, and then filtered with Gblocks. Two corresponding maximum likelihood trees were constructed using RAxML.

**Gene tree incongruence**. To check the (in)congruence of individual tree phylogenies with the recovered trees we performed gene tree analysis. We kept only the alignments of the single-copy groups that included sequence information for all 31 species, and processed them with Gblocks, to keep sites that were aligned properly. The filtered alignments were used to construct individual gene trees using RAxML with -m PROTGAMMAAUTO option for automatic selection of the best fitting model and 100 rapid bootstrap replicates. A majority rule consensus tree was built to summarize the bipartition information of the resulted gene trees using RAxML -J MR option. Also, internode certainty (IC) of each node and the extended internode certainty (ICA) were calculated, as well as the tree certainty (TC) and the extended tree certainty (TCA) values[59]. IC and TC are calculated based on the most prevalent conflicting bipartition, while ICA and TCA take into account all prevalent conflicting bipartitions. Those metrics were calculated using the RAxML -f i option[60] under the $JTT + F + \Gamma 4$ model, which was the one found as optimal during the main ML analysis.

**Statistics and reproducibility**. We used bootstrap and jackknife replicates in the maximum likelihood tree inference framework to estimate the significance of the tree branches. Bootstrap replicates were produced using the RAxML software and jackknife replicates were produced using a custom Python script (github.com/pnatsi/Sparidae_2019/blob/master/jackknife.py).

Various statistical tests to compare our resulting topology to the one in ref. [25] were conducted using the CONSEL tool. All tests and corresponding p-values are reported in Table 4.

No statistical tests were conducted to determine the final number of species included in the analysis.

**Reporting summary**. Further information on research design is available in the Nature Research Reporting Summary linked to this article.

## Code availability

All proteomes, orthogroups and alignments used in this study can be found under https://doi.org/10.5281/zenodo.3250770 and scripts used in this study are available in https://github.com/pnatsi/Sparidae_2019 and https://github.com/pnatsi/PorthoMCL-parser.

## Data availability

The four Sparidae transcriptomes analysed in the present study are available from the corresponding author upon request. Accession numbers for the Sparidae raw sequence reads are: PRJNA395994 (red porgy and common pandora), PRJNA241484 (sharpsnout sea bream) and PRJNA481721 (common dentex). For the gilthead sea bream, we used the predicted gene-set from its genome paper[25]. The genome of gilthead seabream can be accessed through a dedicated genome browser (http://biocluster.her.hcmr.gr/myGenomeBrowser?portalname=Saurata_v1).

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

## Acknowledgements

Financial support for this study has been provided by the Ministry of Education and Religious Affairs, under the Call ARISTEIA I of the National Strategic Reference Framework 2007–2013 (SPARCOMP, #36), co-funded by the EU and the Hellenic Republic through the European Social Fund. This research was supported through computational resources provided by IMBBC (Institute of Marine Biology, Biotechnology, and Aquaculture of the HCMR (Hellenic Centre for Marine Research). Funding for establishing the IMBBC HPC has been received by the MARBIGEN (EU Regpot) project, LifeWatchGreece RI, and the CMBR (Centre for the study and sustainable exploitation of Marine Biological Resources) RI. The authors would also like to sincerely thank Dr Daniel Leite (Department of Genetics, Evolution and Environment, UCL) for his valuable comments regarding the correct use of English language throughout the text.

## Author contributions

P.N. contributed in performing the phylogenetic analyses, producing the figures and drafting the manuscript. A.T. contributed in providing the Sparidae transcriptomes. P.P. contributed in designing the phylogenetic analyses. C.S.T. contributed in conceiving the analysis and writing the manuscript. T.M. contributed in conceiving the analysis, providing the Sparidae transcriptomes, designing and performing the phylogenetic analyses and writing the manuscript. All authors have read and approved the final version of the manuscript.

## Competing interests

The authors declare no competing interests.
