## [Peer Review File · Communications Biology]

Reviewers' comments:

Reviewer #1 (Remarks to the Author):

The manuscript presents an analysis of the phylogenetic position of the sparids. The authors use datasets of 533 and 793 single-copy orthologs from 31 species (26 proteomes + 5 transcriptomes). They show that the Tetraodontiformes are the closest relatives of the sparids, contradicting a previous analysis they had published. This previous study used a smaller set of species and a different method for identifying orthologs. By applying their new method of orthology assignment to the previous dataset they show that they recover the same species tree as obtained in the previous analysis, indicating that the different orthology inference method was not the cause of the different species tree obtained in the new analysis. They suggest instead that taxon sampling may have been the significant factor.

The phylogenetic study appears to be carefully performed and the subsequent tests of the robustness of the phylogenies provide good support for the reported results. The question of why the previous dataset produced a contradictory result has not been resolved.

It would not be particularly easy for a researcher to reproduce the work due to the datasets used for the analysis not being provided e.g. proteomes used for the phylogenomic analysis, orthogroups, alignments.

Major Comments

P4: The authors state that an unrooted phylogeny from Herran et al (2001) divided the sparids into two lineages. An unrooted phylogeny cannot do this--different placements of the root would lead to different hypotheses about the division of the family into lineages. The mis-statement is in the original Herran et al (2001) study but shouldn't be repeated. The division into two clades in that paper only resulted once an assumption of a mid-point root had been made.

P14, "they agreed on the monophyly of the five sparid species". If this claim is to support the quality of the inferred phylogenies then the authors should note in the text that the protein sequences for these five species were all from transcriptomes whereas genomes were used for all the remaining species. The recovered proteomes for these five species are the least complete as shown by the BUSCO analysis and there could be shared patterns in their absence from the orthogroups used for the phylogenetic analysis. Unless steps are taken to ensure no artefacts have arisen from this shared feature between these five taxa then care should be taken in the claims made for their monophyly.

P16: I would suggest performing species tree inference on the 2,192/1,366 genes superalignment from the original study using the CAT model [1], this may help as a possible means of determining the cause of this previous phylogeny contradicting the present one. Standard rate matrices provide averages for amino acid substitutions but often the substitutions at any particular site are actually constrained to just a few possibilities. With low taxon sampling standard rate matrices can create a long branch attraction due to more frequent than expected homoplasies being misidentified as synapomorphies (obviously, this is an over-simplification of how ML actually works but serves to make more intuitive the mathematical effect that results). This could have occurred in this analysis due to the use of only one taxon for the sparids. Using the CAT model may address this and thus resolve the conflict between the two studies.

Minor Comments

I may have missed it but Pauletto et al (2018) does not appear to be in the references. I am assuming this is the same as Pauletto et al (2019), which also appears in the main text but not in the references?

P13 & Fig 3: I think some measure of clustering similarity would be more useful here e.g. RAND index on the orthogroups used to construct the superalignments. It's not obvious from the fact that only 56 were identical whether the orthogroups in these analyses are in fact very different or largely similar.

P14: please could binomial nomenclature be used (or included in parantheses) when referring to placement of species within any of the trees since that is how the species appear in the trees. It is a burden on the reader otherwise.

P15: Similarly the Carangaria (C), Anabantaria (A) and Ovalentaria (O) could be marked on the trees.

P22: "PorthoMCL is the fastest option for genome-scale analyses" - this isn't supported by the PorthoMCL paper, which only has comparisons against OrthoMCL and no other tools. In a quick test of the latest current versions of each, PorthoMCL appeared to be considerable slower than OrthoFinder (for 5 species using 16 cores PorthoMCL took 4 hours versus 15 minutes for OrthoFinder).

References

[1] "Empirical profile mixture models for phylogenetic reconstruction." Le S.Q., Gascuel O., Lartillot N. Bioinformatics. 2008 Oct 15;24(20):2317-23.

Reviewer #2 (Remarks to the Author):

This study hypothesizes the placement of sparids (Teleostei: Spariformes) in the actinopterygian tree using a phylogenomic analysis. The authors used both newly obtained and previously published transcriptome data on five in-group and 26 outgroup taxa. The results infer tetraodontids as the closest relative to sparids. The authors use many various methods pertaining to the effects of taxon sampling and taxa with low support in phylogenomic analyses.

The authors did a wonderful job, I commend them on a well-executed study and a well-written manuscript. They have provided a very robust analysis regarding the placement of the Spariformes. The authors thoroughly address multiple possible influencing factors including taxonomic sampling and long-branch attraction. Although the placement of Spariformes as sister to Tetraodontiformes has been hypothesized previously, I believe the authors have justified their work by using a multitude of tests regarding phylogenetic assessment including gene quality, orthodoxy assessment, and analyses regarding discordance between trees. Both the revised placement of Spariformes since the most recent study and the thorough use of phylogenetic methods will be interesting and informative to readers and may spur future researchers to use similar methods in their phylogenomic analyses.

Comments

1. Figure 3 seems unnecessary to the overall flow of the manuscript, I suggest removing it.
2. Including an image or line drawing of a representative sparid, either as its own figure or incorporated into either the phylogeny or the flowchart, may be helpful and more visually pleasing to readers.

Reviewer #1 (Remarks to the Author):

The manuscript presents an analysis of the phylogenetic position of the sparids. The authors use datasets of 533 and 793 single-copy orthologs from 31 species (26 proteomes + 5 transcriptomes). They show that the Tetraodontiformes are the closest relatives of the sparids, contradicting a previous analysis they had published. This previous study used a smaller set of species and a different method for identifying orthologs. By applying their new method of orthology assignment to the previous dataset they show that they recover the same species tree as obtained in the previous analysis, indicating that the different orthology inference method was not the cause of the different species tree obtained in the new analysis. They suggest instead that taxon sampling may have been the significant factor.

The phylogenetic study appears to be carefully performed and the subsequent tests of the robustness of the phylogenies provide good support for the reported results. The question of why the previous dataset produced a contradictory result has not been resolved.

It would not be particularly easy for a researcher to reproduce the work due to the datasets used for the analysis not being provided e.g. proteomes used for the phylogenomic analysis, orthogroups, alignments.

It is particularly true that, even though we provide accession numbers for every proteome used and detailed description of the methods we applied, reproducing the analysis might prove quite laborious. For that reason, we submitted all proteomes, orthogroups and alignments used in the MS in the zenodo repository. The data consist of 31 fish proteomes, 4 sets of orthogroups (31- and 14-taxa, OrthoFinder and PorthoMCL) and 8 sets of alignments (Gblocks/no Gblocks for each of the 4 orthogroups sets) and can be found under this link: <https://doi.org/10.5281/zenodo.3250770>. To ensure that the readers will be in a position to replicate our work we have also uploaded all scripts used in this study in the code repository [github \(https://github.com/pnatsi/Sparidae_2019\)](https://github.com/pnatsi/Sparidae_2019) and <https://github.com/pnatsi/PorthoMCL-parser>. We have also added these links in the main text (lines 540-543).

Major Comments

P4: The authors state that an unrooted phylogeny from Herran et al (2001) divided the sparids into two lineages. An unrooted phylogeny cannot do this--different placements of the root would lead to different hypotheses about the division of the family into lineages. The mis-statement is in the original Herran et al (2001) study but shouldn't be repeated. The division into two clades in that paper only resulted once an assumption of a mid-point root had been made.

This is a very accurate point and we thank Reviewer #1 for correcting us. We have now changed "unrooted" to "mid-point rooted" and removed the statement about the splitting into two separate lineages (lines 70-71). We still mention however that they found *P. erythrinus* to be closer to *D. dentex* rather than to *P. pagrus*, which is relevant for the rest of the introduction and our results.

P14, “they agreed on the monophyly of the five sparid species”. If this claim is to support the quality of the inferred phylogenies then the authors should note in the text that the protein sequences for these five species were all from transcriptomes whereas genomes were used for all the remaining species. The recovered proteomes for these five species are the least complete as shown by the BUSCO analysis and there could be shared patterns in their absence from the orthogroups using for the phylogenetic analysis. Unless steps are taken to ensure no artefacts have arisen from this shared feature between these five taxa then care should be taken in the claims made for their monophyly.

We agree that transcriptome-based proteomes tend to cluster together and separately from genome-based proteomes, mostly in phylogenetic trees inferred using gene presence/absence information and less often in sequence-based phylogenies. Our approach to treating missing data (keeping orthogroups with at least 27 of the 31 species) ensures that at least 1 of the 5 Sparidae species will be present in every orthogroup alignment, thus possibly accounting for their least complete BUSCO results.

P16: I would suggest performing species tree inference on the 2,192/1,366 genes superalignment from the original study using the CAT model [1], this may help as a possible means of determining the cause of this previous phylogeny contradicting the present one. Standard rate matrices provide averages for amino acid substitutions but often the substitutions at any particular site are actually constrained to just a few possibilities. With low taxon sampling standard rate matrices can create a long branch attraction due to more frequent than expected homoplasies being misidentified as synapomorphies (obviously, this is a over-simplification of how ML actually works but serves to make more intuitive the mathematical effect that results). This could have occurred in this analysis due to the use of only one taxon for the sparids. Using the CAT model may address this and thus resolve the conflict between the two studies.

We thank the reviewer for this proposal. Long-branch attraction (LBA) was the first thing we thought that might have led to the differences between the two datasets as well. However, when we kept only *M. mola* in our current analysis (Supplementary Figure 2C), or the two Tetraodontiformes used in Pauletto et al (*T. rubripes* & *T. nigrovirdis*, Supplementary Figure 2B) we found again the same topology in both cases. In another experimentation, when keeping only *S. aurata* (Supplementary Figure 2A) again we got the same output with using all five sparids. Those findings prompted us to believe that LBA might not play such a big role.

However, it is well-known that accounting for rate heterogeneity suppresses LBA artefacts, which cannot be ruled out for the case of the Tetraodontiformes placement in the Pauletto et al. (2018) tree. Based on this notion, and the Reviewer’s suggestion, we proceeded to initiate two MCMC chains in the program PhyloBayes, using the CAT-GTR model. We used the smallest of the two alignments for this analysis (PorthoMCL, 1366 genes/442,936 sites after Gblocks). The runs were initiated on July 8th, and by July 26th ~300 trees were sampled for each chain using 20 cores from the IMBBC HPC infrastructure. Checking for convergence using ‘bpcomp’ showed a maxdiff of 1, indicating that the runs are far from reliable, suggesting that we would probably need months of computational time to implement this analysis on such a large dataset. However, the topology produced by bpcomp at that point was exactly the same as the one produced by the ML searches conducted with RaxML. For those reasons, we believe that the cost in terms of computational time would be too high in this dataset, especially given the confident results taken from the already conducted analyses.

Finally, to remind the reader of the potential role of LBA in the discordance between the two papers, we have highlighted once again this factor in the discussion of the main text (lines 442-444).

Minor Comments

I may have missed it but Pauletto et al (2018) does not appear to be in the references. I am assuming this is the same as Pauletto et al (2019), which also appears in the main text but not in the references?

Thank you for the correction. This was a typo, we added Pauletto et al (2018) in references (lines 660-661) and changed 2019 to 2018 in the text. We also corrected Tsakogiannis et al. 2019 (line 101)

P13 & Fig 3: I think some measure of clustering similarity would be more useful here e.g. RAND index on the orthogroups used to construct the superalignments. It's not obvious from the fact that only 56 were identical whether the orthogroups in these analyses are in fact very different or largely similar.

We decided to completely remove Fig. 3 from the analysis, as suggested by Reviewer #2. It is true that the way it was made did not provide any insightful information to the reader. We also removed the reference to this figure from within the text (line 275)

P14: please could binomial nomenclature be used (or included in parentheses) when referring to placement of species within any of the trees since that is how the species appear in the trees. It is a burden on the reader otherwise.

We changed every reference to a particular species to latin names and common names were put in parentheses, together with the family that the species belongs to (lines 291-298, 303-306, 319, 325, 344-346, .

P15: Similarly the Carangaria (C), Anabantaria (A) and Ovalentaria (O) could be marked on the trees.

The series of the fish groups, as described by Betancur-R et al. (2014) are now indicated on the corresponding branches in the relevant trees.

P22: "PorthoMCL is the fastest option for genome-scale analyses" - this isn't supported by the PorthoMCL paper, which only has comparisons against OrthoMCL and no other tools. In a quick test of the latest current versions of each, PorthoMCL appeared to be considerable slower than OrthoFinder (for 5 species using 16 cores PorthoMCL took 4 hours versus 15 minutes for OrthoFinder).

Thank you for the correction. We changed "... PorthoMCL is the fastest option for genome-scale analyses." to "... PorthoMCL is a faster implementation of the OrthoMCL algorithm." (lines 485-486)

=====

Reviewer #2 (Remarks to the Author):

This study hypothesizes the placement of sparids (Teleostei: Spariformes) in the actinopterygian tree using a phylogenomic analysis. The authors used both newly obtained and previously published transcriptome data on five in-group and 26 outgroup taxa. The results infer tetraodontids as the closest relative to sparids. The authors use many various methods pertaining to the effects of taxon sampling and taxa with low support in phylogenomic analyses.

The authors did a wonderful job, I commend them on a well-executed study and a well-written manuscript. They have provided a very robust analysis regarding the placement of the Spariformes. The authors thoroughly address multiple possible influencing factors including taxonomic sampling and long—branch attraction. Although the placement of Spariformes as sister to Tetraodontiformes has been hypothesized previously, I believe the authors have justified their work by using a multitude of tests regarding phylogenetic assessment including gene quality, orthology assessment, and analyses regarding discordance between trees. Both the revised placement of Spariformes since the most recent study and the thorough use of phylogenetic methods will be interesting and informative to readers and may spur future researchers to use similar methods in their phylogenomic analyses.

Comments

1. Figure 3 seems unnecessary to the overall flow of the manuscript, I suggest removing it.

We removed Fig. 3 and the corresponding paragraph of the manuscript. It is true that the way it was made did not provide any insightful information to the reader. Fig. 4 from the initial submission is now labeled as Fig. 3

2. Including an image or line drawing of a representative sparid, either as its own figure or incorporated into either the phylogeny or the flowchart, may be helpful and more visually pleasing to readers.

Images of two from the five sparids used in our study now accompanies the flowchart of the analysis in Fig. 1. The images were obtained from one of the authors' (AT) personal work .

REVIEWERS' COMMENTS:

Reviewer #1 (Remarks to the Author):

I'm happy to confirm that the authors have addressed the questions I raised in the first review.

Reviewer #2 (Remarks to the Author):

This study regarding the phylogenetic analysis of sparids using transcriptome data is both thorough and well written. The authors' inclusion of multiple tests regarding the robustness and influencing factors on taxon placement is refreshing to read. The authors have thoughtfully responded to all reviewer comments, and after reading through the revised manuscript I have no further concerns.